# Artificial Intelligence in NICU and PICU: A Need for Ecological Validity, Accountability, and Human Factors

**DOI:** 10.3390/healthcare10050952

**Published:** 2022-05-21

**Authors:** Avishek Choudhury, Estefania Urena

**Affiliations:** 1Industrial and Management Systems Engineering, West Virginia University, Morgantown, WV 26506, USA; 2Registered Nurse, Intensive Critical Unit, Lincoln Medical and Mental Health Centre, New York, NY 10451, USA; estefania.urena@nychhc.org

**Keywords:** Artificial Intelligence, technology readiness level, accountability, reliability, liability, workload, pediatric

## Abstract

Pediatric patients, particularly in neonatal and pediatric intensive care units (NICUs and PICUs), are typically at an increased risk of fatal decompensation. That being said, any delay in treatment or minor errors in medication dosage can overcomplicate patient health. Under such an environment, clinicians are expected to quickly and effectively comprehend large volumes of medical information to diagnose and develop a treatment plan for any baby. The integration of Artificial Intelligence (AI) into the clinical workflow can be a potential solution to safeguard pediatric patients and augment the quality of care. However, before making AI an integral part of pediatric care, it is essential to evaluate the technology from a human factors perspective, ensuring its readiness (technology readiness level) and ecological validity. Addressing AI accountability is also critical to safeguarding clinicians and improving AI acceptance in the clinical workflow. This article summarizes the application of AI in NICU/PICU and consecutively identifies the existing flaws in AI (from clinicians’ standpoint), and proposes related recommendations, which, if addressed, can improve AIs’ readiness for a real clinical environment.

## 1. Artificial Intelligence in Pediatrics

With increasing healthcare infrastructure and connected medical databases, clinicians have more data to inform clinical decision-making than ever before. However, when confronted with information beyond the scope of their expertise and in excessive quantities, they are likely to resort to boundedly rational and, in some cases, incorrect diagnoses. One way to support complex clinical processes is to leverage Artificial Intelligence (AI) technologies, often known as AI-based clinical decision support systems. As portrayed by the media, AI comes with surprising capabilities in healthcare. AI can be broadly defined as an intelligent system capable of performing human-like activities based on retrospective data. A typical AI system encompasses predefined rules, if–then statements, or is powered by dynamic statistical models that are proficient in capturing non-linear relationships among several variables. More recently, wide arrays of unique AI technologies have been developed to augment the healthcare system. The US Food and Drug Administration (FDA) has approved several AI-based products, signaling the gradual integration of AI into healthcare [1,2].

Pediatric patients are typically at an increased risk of fatal decompensation and are sensitive to medications. That being said, any delay in treatment or minor errors in medication dosage can overcomplicate patient health. Under such an environment, clinicians are expected to quickly and effectively comprehend large volumes of medical information to diagnose and develop a treatment plan for a given baby. Being one of the most complex and sensitive healthcare domains, neonatal and pediatric intensive care units (NICUs and PICUs) are ideal environments for AI use, where doctors and nurses can leverage AIs’ computational capabilities to make well-informed and faster clinical decisions. The use of AI in pediatrics was first recorded in 1968 when Paycha developed SHELP, a computer-assisted medical decision-making system that diagnosed inborn errors of metabolism [3]. Soon after, Shortliffe developed an expert system named Mycin and identified bacteria causing severe blood infections among pediatric patients [4]. Since then, as AI has developed, several randomized controlled trials have used the technology for various issues in pediatrics. For instance, a study implemented an automated AI-based decision support system to control glucose levels effectively and safely among pediatric patients [5]. Another study developed an AI-based wearable device known as the Superpower Glass to augment the social outcomes of children with autism [6]. A study conducted in China successfully developed an AI-based disease risk prediction model for newborn babies with inherited metabolic diseases [7]. A study reported a significant improvement in neurocognitive performance among children when an AI-based cognitive stimulation therapy was implemented [8]. Besides clinical trials, several other AI technologies have been developed that play an active role in neonatal and pediatric ICUs. For example, AI-based models have been used in the NICU to predict birth asphyxia [9,10] and neonatal seizures [11], as well as to diagnose neonatal sepsis [12,13] and respiratory distress syndrome [14]. Table 1 gives a snapshot of the various applications of AI in NICU and PICU.

Overall, different studies have used AI either to directly improve patient health by allowing physicians “spend more time in direct patient care [while reducing provider burnout]” [23] or to augment clinical processes thus improving patient health indirectly. For instance, a study conducted in California reported AI’s efficiency in identifying critically ill PICU patients with an underlying genetic disorder [24]. A study in Spain used AI-driven music to reduce stress levels among neonates [25]. Several studies used AI algorithms to develop an early warning system that provided a timely detection of changes in health status and the development of critical illness [12,15,17,19,26,27,28] and pathologic eye disease progression in preterm infants [29]. A recent review also reported several ‘indirect impacts’ of AI on the pediatric patient [30,31], where AI was noted to augment clinical decision-making and diagnostic accuracy in the pediatric setting [21,24,28].

## 2. Current Challenges Preventing AI Application

Despite all the evidence supporting AI in pediatrics, its use and adoption have been limited. Even though no studies thus far have associated AI with worsened health outcomes or patient harm in a pediatric, *why do doctors and healthcare management hesitate to integrate AI into their clinical workload?* Of all possible reasons hindering the acceptance of AI in pediatrics, **(a)** the lack of ecological validity, and **(b)** low technology readiness level, two inter-related factors, along with the **(c)** lack of AIs’ accountability, seem to be prominent determinants that have not been sufficiently acknowledged in the literature.

### 2.1. Ecological Validity—Can the User Use AI Effectively and Safely?

As depicted in several studies, AI systems and technologies may facilitate a personalized approach to pediatric care by augmentation of diagnostic processes. The AI-based solution has the power to reinvigorate clinical practices. Although the advent of personalized patient treatment is provocative and often crucial in a pediatric environment, there is a need to assess the true potential of AI when implemented in a real, uncontrolled, and chaotic healthcare scenario. In all the studies published around this topic, the experiments were either conducted retrospectively or by experts in a controlled setting, therefore lacking ecological validity. Recent systematic reviews [32,33] analyzing AIs’ role and performance in healthcare acknowledged that AI systems or models were often evaluated under unrealistic conditions (controlled research environment) that had minimal relevance to routine clinical practice (workload, chaos, and time constraints). Therefore, there is a lack of evidence exhibiting AIs’ efficacy in a real clinical environment.

It is essential to understand that the working environment and cognitive workload are significant determinants of technology use. In a pediatric setting, clinicians are often assigned several patients with unique needs and health statuses. Given the global shortage of staff and the increasing burden on the healthcare industry, clinicians often experience burnout and fatigue. Individuals under such stress and discomfort might not be efficient in utilizing AI devices and comprehending its outcome in the same way as reported in several research articles. Therefore, studies must evaluate AI systems under a real scenario to ensure effective use when integrated into a clinical workflow.

### 2.2. Technology Readiness Level

Recently, several innovations around medical AI have been associated with excellent performance in the literature. However, research breakthroughs do not necessarily translate into a technology that is ready to use in a high-risk environment such as healthcare [32,33]. That said, most AIs featuring prominent abilities in research and literature, for the most part, would not be executable in a clinical environment. According to the Technology Readiness Level (TRL), most AI systems, at least in pediatric and neonatal intensive critical care (PICU and NICU), if not all, do not qualify for implementation. TRL is a gauging system developed to assess the maturity level of a particular technology [34]. TRL consists of nine categories (readiness levels), where a score of TRL 1 is the lowest, and TRL 9 is the highest (**see Box 1**). By applying the TRL system to the articles involving AI in pediatrics, we can observe that most published articles are prototype testing in an operational environment with near-implementation readiness (TRL 7). Few to none of the AI systems discussed in the literature have been deployed into a real ICU setting and evaluated longitudinally over a significant duration.

Box 1Technology Readiness Levels (1–9).Technologies with TRL 1 through 4 are executable in laboratory setting, where the main object is to conduct research. This stage is the proof of concept.
-TRL 1: Basic principles of the technology observed-TRL 2: Technology concept formulated-TRL 3: Experimental proof of concept developed-TRL 4: Technology validated in a study laboratoryTechnologies with TRL 5 through 7 are in the development phase, where the functional prototype is ready.
-TRL 5: Technology validated in relevant environment (controlled setting in a real-life environment)-TRL 6: Technology demonstrate in relevant environment-TRL 7: System prototype demonstrated in operational environmentLastly, technologies with TRL 8 and 9 are in the operational phase where the primary objective is implementation.
-TRL 8: System completed and certified for commercial use

### 2.3. AI Accountability—Who Is Responsible for Technology Error?

How does the absence of AIs’ accountability impact clinicians’ intention to use the technology? This chapter explains ‘accountability’ as a process in which healthcare practitioners have potential responsibilities to justify their ‘clinical actions’ to patients (or families) and are held liable for any impending positive or negative impact on patient health. While using an AI-based decision support system, only clinicians are held accountable if they decide to follow an AI-based treatment, resulting in patient harm. Additionally, clinicians are also held responsible if they deviate from the standard protocols [1]. This may be worrisome because, under such circumstances, clinicians will only follow AI if it matches their judgment and aligns with the standard protocol—making the AI underused.

Furthermore, it might be difficult for clinicians, who are not necessarily trained in the subject, to effectively comprehend AIs’ functioning under an existing burnout state and identify any technological flaw. One way to address the problem of ‘accountability’ is by training doctors and nurses to understand when to rely upon or not on AI recommendations. However, training or educating practitioners on AI will require substantial effort. The AI accountability issue solution will require a systematic approach involving stakeholders from the law, policymakers, computer scientists, human factors researchers, healthcare organizations, healthcare practitioners, insurance agencies, and patients.

## 3. Recommendations and Future Steps

Concerns regarding Ecological Validity and TRL can be associated with AIs’ usability. There is a lack of studies evaluating the usability or user-centeredness of any AI technology in a pediatric setting. As acknowledged earlier in this chapter, clinicians are often overwhelmed with clinical responsibilities. Therefore, to ensure the adoption of AI in pediatrics, it is essential to develop systems that are easy to use and that fulfill pediatric nurses’ and doctors’ requirements. AI developers also need to consider the end-user of their products. Since most bedside tasks are performed by nurses, the AI system implemented at the bedside should be designed for nurses, as their digital literacy can be substantially different from other physicians or researchers (study participants) and may vary across demographics.

Future studies should include pediatric populations with multiple chronic complexities in randomized controlled trials. Current approaches to pediatric AI usually emphasize single diseases, which may have minimal relevance to a real complex scenario. Another consideration is to have an adaptive algorithm that can gauge patients’ health status and evolve over time. Therefore, future research efforts to integrate AI systems into pediatric settings need to match the measure and underlying disease trajectory to patients’ situations.

Until now, all studies have been focused on the patient. What’s missing in the literature is the use of AI to address clinicians’ concerns. Addressing clinicians’ problems can not only improve their clinical performance but also augment care quality. The pediatric unit (PICU and NICU) is one of the most critical departments within any healthcare establishment. For example, while dealing with a pediatric patient, particularly in a NICU or PICU setting, the clinicians need to consider the body size differences between every pediatric patient and consecutively be aware of all the continuous physical and cognitive development of their patients. That being said, the medication dosage (which largely depends on the body weight) might change over time for a pediatric patient (depending on their rate of physical growth). Additionally, clinicians need to have special consideration while intubating pediatric patients as they have larger tongues and a uniquely positioned epiglottis and larynx. Pediatric patients also have subtle cardiovascular differences, making heart rate a critical clinical factor. They are also prone to pathogens and neurological disorders from poisoning. In other words, pediatric patients have a very low tolerance to any error, and therefore, clinicians are required to pay for extra care and personalized treatment.

Apart from caring for patients, pediatric clinicians also have to dedicate a significant amount of time and effort to educating patients’ parents. Such a work demand often takes a heavy toll on their cognitive workload, and AI technologies can be developed to identify clinicians undergoing excessive cognitive load or burnout. Since clinicians in a burnout state are prone to human errors, identifying and providing them with timely assistance can help ensure patient safety. Identifying cognitive workload will also help the floor manager to better schedule their staff and designate appropriate resources.

Night nurses, particularly those who are new in the profession, may feel exhausted during their shifts. In a setting where nurses have to keep a continuous watch on patient monitors (a critical aspect in NICU and PICU settings), performing efficiently often becomes challenging. In such a scenario, AI, in conjunction with eye trackers, can be leveraged to measure nurses’ attention span and to identify the zone in the screen where they gaze. AI can then optimize the information being displayed on the clinical monitors to highlight the essential data in real-time.

AI technology can be used to identify and record clinicians’ behavior leading to near misses so that it can generate an alert in the future. It is essential to acknowledge that in healthcare, outcomes are reasonable because clinicians make educated and just-in-time adjustments according to the fluctuating health condition. Future work should train AI on the critical adjustments made by clinicians, so that AI can adapt in real-time in the same manner as experienced clinicians do. Please note that the views present in this article can differ from those of experts in AI and across different healthcare settings; hence it should be considered with caution. 

## 4. Major Takeaways

Artificial Intelligence has great potential, but the consideration of human factors is essential for its sustainability in pediatrics.The lack of AIs’ ecological validity hinders its adoption and usage in the clinical workflow.The lack of AIs’ accountability can be a significant hurdle in AI acceptance among clinicians.Artificial Intelligence, if used appropriately, can improve clinical workflow and, in turn, augment the quality of care.All AI-based decision support systems should be exclusively designed for their end-users (doctors and nurses) to safeguard the technology as well as patient safety.

## Figures and Tables

**Table 1 healthcare-10-00952-t001:** State of the art: Artificial Intelligence in PICU and NICU (not an exhaustive list).

Study	Institution(s)	Patients	Data Source and Type	Model	Compared with Clinicians	Conclusion
[15]	Autism Brain Imaging Data Exchange Database	28	Research database: Images	Artificial Neural Network	No	The study accurately predicted cognitive deficits/function in individual very preterm infants soon after birth. However, larger data size is required to achieve the clinical gold standard.
[16]	Italian Neonatal Network	23,747	Research database: Numerical	Artificial Neural Network	No	The study shows that using the only limited information available up to 5 min after birth. AI can have a significant advantage over current approaches in predicting the survival of preterm infants.
[17]	German Tertiary Care PICU	296	EHR: Numerical	Random Forest	No	The study shows that AI can facilitate the early detection of sepsis with an accuracy superior to traditional biomarkers. It can also potentially reduce antibiotic use by 30% in non-infectious cases.
[18]	Cambridge University	94	EHR: Numerical	Support Vector Machine	No	The study shows how AI algorithms can predict severe traumatic injury outcomes at six months using just the three most informative parameters.
[19]	Severance Hospital and Samsung Medical Center	1723	EHR: Numerical	Convolutional Neural Network	No	The study demonstrated that the machine learning-based model, the Pediatric Risk of Mortality Prediction Tool, can outperform the conventional Pediatric Index of Mortality scoring system in predictive ability.
[20]	University Hospital EHR	93	EHR: Numerical	Naïve Bayesian models	Yes	The study demonstrates the capability of AI models in augmenting clinicians’ ability to identify infants with single-ventricle physiology at high risk of critical events. The study also reports that the early prediction of critical events may improve the overall care quality and minimize health care expenses.
[21]	University of Pittsburgh	37	Research database: EEG signals	Long Short-Term Memory	No	The algorithm proposed in the study gave promising results in automatic sleep stage scoring in neonatal sleep signals.
[22]	St. Louis Children’s Hospital	285	EHR: Numerical	Novel Deep Learning Model	No	The novel AI model developed in the study demonstrated efficacy in predicting the real-time mortality risk of preterm infants in initial NICU hospitalization. The proposed model also outperformed the existing clinical risk index II scoring system for babies

EHR = electronic health records; EEG = electroencephalogram; AUROC = area under the receiver operating characteristic curve.

## Data Availability

Not applicable.

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
