# Peer review of "Artificial Intelligence in NICU and PICU: A Need for Ecological Validity, Accountability, and Human Factors"

_healthcare, 2022, doi:10.3390/healthcare10050952_

Round 1

Reviewer 1 Report

In the abstract section, authors name AI without explaining that it is short for Artificial Intelligence. 

The authors mention in Table 1 some topics that are not explained before, as for example: models, classification type, cross validation, machine learning, accuracy and AUROC. These concepts should be described to contextualise the reader. 

Papers mentioned should be revised. For example, He L, (15) in table 1, authors indicate the IA model used is SVM, however, He L, (15) propose an ANN framework with a stack of models.

The paragraph below table 1 names papers that are not included in the table and viceversa.

Author Response

In the abstract section, authors name AI without explaining that it is short for Artificial Intelligence. 

Response: Thank you for the comment, We have fixed the error.

The authors mention in Table 1 some topics that are not explained before, as for example: models, classification type, cross validation, machine learning, accuracy and AUROC. These concepts should be described to contextualize the reader. 

Response: We understand your concern. However, to avoid any confusion or to deviate from the main focus, we deleted the technical columns from the table. The remaining content is intended for non-AI readers. (potential end users)

Papers mentioned should be revised. For example, He L, (15) in table 1, authors indicate the IA model used is SVM, however, He L, (15) propose an ANN framework with a stack of models.

Response: Thank you for identifying this error. We have rectified this.

The paragraph below table 1 names papers that are not included in the table and viceversa.

Response: The table only gives a snapshot of the state of the art where AI was implemented in NICU or PICU whereas in the text we explain the use of AI (at the research stage) in any pediatric setting.

Reviewer 2 Report

First of all congratulations for the research. I've found it a really interesting lecture. I appreciate the effort that authors dedicated to preparing the manuscript about utilization of artificial intelligence (AI) in the form of computer aided diagnosis and decision support systems in pediatrics. Although AI is very promising in this field, it is under-exploited due to limitations in ecological validity and technology readiness level (TRL).

The article is clear and straightforward, and the take-home messages are presented clearly. English utilization is adequate. I have only some minor concerns reported in the folllowing.

ARTIFICIAL INTELLIGENCE IN PEDIATRICS
I think there is a type error (babe for babies) in page 2, line 10: "model for newborn babes with inherited metabolic diseases".
The authors report both studies presenting Decision Support Systems (DSS) and Computer Aided Diagnosis (CAD) in this paragraph; however, much enphasis is given to DSS and none to CAD. Please briefly highlight the utilization of AI as a CAD, as well as you did with DSS.

TECHNOLOGY READINESS LEVEL
As the authors reported a box with the TRL levels, it would help if they stated explicitly in the text  what TRL they refer to when they write "we can observe that most published articles are prototype testing in an operational environment with near-implementation readiness." 

I also have some concerns regarding the references included.
- reference 3 (Paycha 1968): the authors should double-check the year, as the text reports 1986 (page 2, line 1), whereas the reference reports 1968.
- reference 5 (Nimri 2020): this study presents an interesting clinical trial on AI-based insulin therapy adjustments for youths with diabetes. However, from the AI point of view, that paper presents a limitation, because the AI algorithm is not described "due to proprietary reasons". To this reviewer's knowledge, some works exist that  describe clearly the Machine/Deep Learning methodology utilized, and that focus specifically on the prediction of blood glucose concentration in children as a DSS. Although they are not primarily used in ICU, in my opinion the article could benefit if the existence of such systems was briefly cited and discussed (for example, do they resort to edge computing or to Internet of Things? Is one of the two approaches better for the clinicians?). One or two of such articles could be added/substituted to the reference of Nimri et al. to enrich the paper. 
- references 9 and 10: I think the word "editor" between the authors and the title is a typo. Please double-check the references.
- references 23 and 24: they have the same title. Are they different? Are both the references necessary? Also, reference 24 has the same problem as ref. 9 and 10.

Again, good work.

Author Response

First of all congratulations for the research. I've found it a really interesting lecture. I appreciate the effort that authors dedicated to preparing the manuscript about utilization of artificial intelligence (AI) in the form of computer aided diagnosis and decision support systems in pediatrics. Although AI is very promising in this field, it is under-exploited due to limitations in ecological validity and technology readiness level (TRL).

Response: Thank you for the acknowledgment

The article is clear and straightforward, and the take-home messages are presented clearly. English utilization is adequate. I have only some minor concerns reported in the folllowing.

ARTIFICIAL INTELLIGENCE IN PEDIATRICS
I think there is a type error (babe for babies) in page 2, line 10: "model for newborn babes with inherited metabolic diseases".

Response: Thank you for the detailed comment. We have proofread the manuscript and fixed the error.

The authors report both studies presenting Decision Support Systems (DSS) and Computer Aided Diagnosis (CAD) in this paragraph; however, much enphasis is given to DSS and none to CAD. Please briefly highlight the utilization of AI as a CAD, as well as you did with DSS.

Response: We agree and understand your concern. The reason for focusing on decision support systems is because usually CDSS(s) are implemented in a clinical setting. In this paper we are interested in identifying the limitation of AI in a clinical setting and not only in the research stage, we focus on CDSS. Also, the implementation of AI-based CAD in NICU and PICU is very limited and is captured in the table for reference.

TECHNOLOGY READINESS LEVEL
As the authors reported a box with the TRL levels, it would help if they stated explicitly in the text  what TRL they refer to when they write "we can observe that most published articles are prototype testing in an operational environment with near-implementation readiness." 

Response: We have mentioned the TRL level in the text. TRL 7

I also have some concerns regarding the references included.
- reference 3 (Paycha 1968): the authors should double-check the year, as the text reports 1986 (page 2, line 1), whereas the reference reports 1968.

Response: Thank you. We have fixed the typo.

- reference 5 (Nimri 2020): this study presents an interesting clinical trial on AI-based insulin therapy adjustments for youths with diabetes. However, from the AI point of view, that paper presents a limitation, because the AI algorithm is not described "due to proprietary reasons". To this reviewer's knowledge, some works exist that  describe clearly the Machine/Deep Learning methodology utilized, and that focus specifically on the prediction of blood glucose concentration in children as a DSS. Although they are not primarily used in ICU, in my opinion, the article could benefit if the existence of such systems was briefly cited and discussed (for example, do they resort to edge computing or to the Internet of Things? Is one of the two approaches better for the clinicians?). One or two of such articles could be added/substituted to the reference of Nimri et al. to enrich the paper. 

Response: Thank you for suggesting the article. It was an interesting read. However, to stay within the scope of this article, we could not include it. However, we will consider it in our upcoming study that encompasses pediatrics in general.

- references 9 and 10: I think the word "editor" between the authors and the title is a typo. Please double-check the references.

Response: We have fixed the typo.

- references 23 and 24: they have the same title. Are they different? Are both the references necessary? Also, reference 24 has the same problem as ref. 9 and 10. Again, good work.

Response: They have the same title but different authors and different papers led by same first author

Reviewer 3 Report

It seems that the authors have a different view from those of experts in artificial intelligence (AI) on the strengths and weaknesses of AI. As an expert in AI, I would like to describe our general perspective on this issue. AI is essentially a data-driven approach. It is NOT much about experimental design (e.g., randomized controlled trial). Instead, it is much about how to collect big data with high quality.

The successful collection of big data with high quality secures the successful training and validation of AI, which in turn leads to its successful application in real-world situations. I would like to strongly suggest the authors to address that their view can be different from those of experts in AI hence it should be considered with caution. 

AI experts pay a lot of attention to the issue of external validation. Here, external validation means evaluating the validity of AI for external data which do not involve the training of AI at all. For improving the external validation of AI, we need to use complicated and inconsistent data for training AI, given that real-world data are expected to be as such. However, the authors suggest the randomized control trial as an ideal source of data collection, even though real-world data would be much more complicated and much less consistent than those from the randomized control trial. The authors should address this critical issue in great detail in the section of Discussion. 

Author Response

It seems that the authors have a different view from those of experts in artificial intelligence (AI) on the strengths and weaknesses of AI. As an expert in AI, I would like to describe our general perspective on this issue. AI is essentially a data-driven approach. It is NOT much about experimental design (e.g., randomized controlled trial). Instead, it is much about how to collect big data with high quality. 

Response: We absolutely agree. However, in order to improve AI acceptance, a clinical and human factors perspective is critical without which the true potential of AI will not be realized. This problem has been acknowledged widely by different authors.

The successful collection of big data with high quality secures the successful training and validation of AI, which in turn leads to its successful application in real-world situations. I would like to strongly suggest the authors to address that their view can be different from those of experts in AI hence it should be considered with caution. 

Response: We agree and have acknowledged it in the manuscript. AI can be trained and improved with more data. But if clinicians are not willing to use it, then the effort shall go in vain.

AI experts pay a lot of attention to the issue of external validation. Here, external validation means evaluating the validity of AI for external data which do not involve the training of AI at all. For improving the external validation of AI, we need to use complicated and inconsistent data for training AI, given that real-world data are expected to be as such. However, the authors suggest the randomized control trial as an ideal source of data collection, even though real-world data would be much more complicated and much less consistent than those from the randomized control trial. The authors should address this critical issue in great detail in the section of Discussion. 

Response: We understand your concern. However, when we see critical settings such as NICU and PICU where patients have a rare disease and big databases are not available for them. In that context, clinical trials would be beneficial. Again, we are not advocating for a general AI for the entire pediatric setting but to have special systems tailored for a select population.

Reviewer 4 Report

The manuscript consists of total 9 pages, including 1 table, 1 figure (box) and the list of total 37 literature references. The manuscript presents as a concise review concerning the possibilities and limitations of using artificial intelligence-based solutions as aids in solving clinical decisions in urgent cases in neonatology and pediatric intensive care. As such, the manuscript fits into the scope of works published in the Journal. The Title is relevant to the contents of the manuscript but a bit too enigmatic; maybe adding some verb to indicate the suggested action to be taken would help. The text has a logical structure, but not the typical and widely accepted structure of scientific article (introduction-material and methods - results - discussion - conclusion), and is written in high quality English.

The Abstract is not structured and narrative but it mirrors the crucial contents of the main text adequately.

In the page 1 section 1 the definition is looped: "AI [Artificial INTELLIGENCE can be broadly defined as an INTELLIGENT SYSTEM..." - it would be tempting to define what the "intelligence" meaning is in the AI context.

I would suggest to stress a bit more and discuss in more detail the problem of accountability for the effects of using  the AI guidance by physicians (or for not using it - in cases when it is available or even contradicts the physician's opinion based on the established clinical knowledge), which the Author raised in the end of the manuscript - as it presents to me as the crucial obstacle in the way towards wider acceptance of AI-based solutions in medicine. 

The literature referred is numerous and recent enough, relevant to the topic of the article. 

Nevertheless, the Authors may consider mentioning the following aspects in the article:
- theoretic aspects of machine learning use in the medical field, as in e.c. https://doi.org/10.3390/app12094649 https://doi.org/10.3390/app12094455 https://doi.org/10.3390/diagnostics12051042
- especially focusing on medical emergencies, as in e.c. https://doi.org/10.3390/electronics11091507 https://doi.org/10.3390/brainsci12050612 https://doi.org/10.3390/jpm12050700 
- and the generally arising ethical dilemmas resulting from AI introduction into practice as in e.c. https://doi.org/10.3390/app12094130 https://doi.org/10.3390/philosophies7030046 

Author Response

The manuscript consists of total 9 pages, including 1 table, 1 figure (box) and the list of total 37 literature references. The manuscript presents as a concise review concerning the possibilities and limitations of using artificial intelligence-based solutions as aids in solving clinical decisions in urgent cases in neonatology and pediatric intensive care. As such, the manuscript fits into the scope of works published in the Journal. The Title is relevant to the contents of the manuscript but a bit too enigmatic; maybe adding some verb to indicate the suggested action to be taken would help. The text has a logical structure, but not the typical and widely accepted structure of scientific article (introduction-material and methods - results - discussion - conclusion), and is written in high quality English.

Response: Being an opinion paper, we do not have the same structure that of a research article.

The Abstract is not structured and narrative but it mirrors the crucial contents of the main text adequately.

Response: Being an opinion paper, we do not have the same structure that of a research article. In this article, unstructured abstract fits better.

In the page 1 section 1 the definition is looped: "AI [Artificial INTELLIGENCE can be broadly defined as an INTELLIGENT SYSTEM..." - it would be tempting to define what the "intelligence" meaning is in the AI context.

Response: AI definition is well defined in the literature and defining “intelligence” separately is not typical or common practice in this field.

I would suggest to stress a bit more and discuss in more detail the problem of accountability for the effects of using the AI guidance by physicians (or for not using it - in cases when it is available or even contradicts the physician's opinion based on the established clinical knowledge), which the Author raised in the end of the manuscript - as it presents to me as the crucial obstacle in the way towards wider acceptance of AI-based solutions in medicine. 

Response: We agree and have expanded on the topic of accountability.

The literature referred is numerous and recent enough, relevant to the topic of the article. 

Nevertheless, the Authors may consider mentioning the following aspects in the article:

Response: Thank you for suggesting interesting articles. However, we are only considering AI in NICU and PICU. However, these articles will be a great fit for our upcoming papers.

Round 2

Reviewer 3 Report

I believe you revised the manuscript well based on my comments and it is ready for publication. I am really grateful about it.